# THE TURING GAME 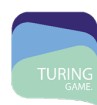

## ABSTRACT

We present first experimental results from the *Turing Game*, a modern implementation of the original imitation game as proposed by Alan Turing in 1950. The Turing Game is a gamified interaction between two human players and one AI chatbot powered by state-of-the-art Large Language Models (LLMs). The game is designed to explore whether humans can distinguish between their peers and machines in chat-based conversations, with human players striving to identify fellow humans and machines striving to blend in as one of them. To this end, we implemented a comprehensive framework that connects human players over the Internet with chatbot implementations. We detail the experimental results after a public launch at the Ars Electronica Festival in September 2024. While the experiment is still ongoing, in this paper we present our initial findings from the hitherto gathered data. Our long term vision of the project is to deepen the understanding of human-AI interactions and eventually contribute to improving LLMs and language-based user interfaces.

## 1 INTRODUCTION

AI systems are built with the goal of performing activities that were traditionally reserved to humans, from playing strategy games, like chess (Campbell et al., 2002), Go (et al., 2016) or Dota-2 (et al., 2019), to generating artistic imagery (Mid) and written texts (OpenAI, 2023; Jones and Bergen, 2023). They became better and better up until the point where some have already surpassed human performances in fields that have traditionally been believed to require human abstract thinking and strategic planning. In the field of content generation, we have arrived at the point where we find it hard to discern whether images or clips are generated or represent real footage or whether texts stem from a human or a machine.

Alan Turing, a founding figure in computer science, posed the question of whether machines can think (Turing, 1950; Saygin et al., 2000). Drawing from the theoretical Turing Machines - capable of computing anything computable - and emerging insights into brain function, Turing hypothesized that human thought processes could be computationally replicated. To address the ambiguous concept of "thinking", he proposed the *Turing Test* (also called "Imitation Game"), where an interrogator communicates with a machine and a human through a text interface, aiming to distinguish between them. The machine passes if it convinces the interrogator it is human, suggesting it simulates aspects of human thought. However, the test's outcomes depend heavily on the participants' motivations and susceptibility to deception, issues unaddressed in Turing's original formulation or later implementations like the Löbner Prize (Shieber, 1994; Epstein et al., 2008) and "Human or Not?" (Jannai et al., 2023).

In this paper, we propose to extend the Imitation Game by symmetrizing the roles of the original two human participants, see Fig. 1. This seemingly slight redesign of the test shifts the focus away from the simple question-answering to the collaboration between the humans and the inference of their mutual intentions, a characteristic feature of human communications (Tomasello et al., 2005; Zhang et al., 2024). Due to that, the question comes down to which of the interlocutors understands the intentions better, a human or a machine. Note that in this way we avoid being explicit about what behavior is *human-like*, allowing participants to decide what behavior is human-like, and what is not. Just like Alan Turing, we leave the kind and length of the conversation fully up to the humans.

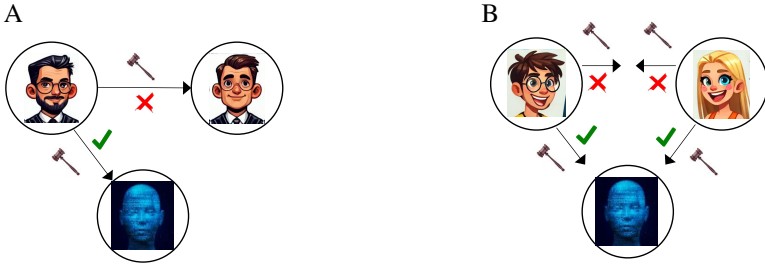

Figure 1: **A** The original Turing Test (Imitation game): The judge has to decide whom of the other two interlocutors he thinks to be the machine. The other human serves as the counterpart to the machine. His role is solely to support the judge in his decision. **B** Our Turing Game: Both humans independently decide which interlocutor they believe is the machine while supporting the other human. Red crosses show a human misidentifying another human as a machine, while green checks indicate correct identification of the machine. Hammers indicate the decision-making. Only if both humans are correct, they win the game.

Humans often express their verbalized thoughts in a non-explicit and incomplete way (Clark and Brennan, 1991). In order for a machine to correctly understand human desires and needs, it needs to understand our thoughts on a large enough joint context (common knowledge), and thus behave as human-like as possible (Christian, 2020; Amodei et al., 2016). Our contributions are as follows:

- We propose a generalization of the Turing Test, the *Turing Game*, which is symmetric with respect to the role of the two humans. We also develop a tailored matching algorithm to pair human players according to their playing performance and average time to make decisions.

- We have developed and installed the Turing Game as a platform and made it publicly available.[1] Our platform serves as a sandbox for testing various LLMs and chatbot implementations designed to mimic human-like thinking, evaluated by an open community. We have designed the ratings of the bots such that the most qualified humans contribute the most to those ratings.

- We present the preliminary experimental results from the hitherto gathered data, mainly from a public exposition and public installation at the Ars Electronica Festival.

The paper is organized as follows: Sec. 2 details the related work and shortcomings of hitherto implementations of Turing-like tests; Sec. 3 describes the proposed Turing Game; Sec. 4 presents results and their analysis from the already gathered data; Sec. 5 concludes and reflects on our contributions. In the Appendix, Sec.A discusses potential ethical consequences, Sec.B complements the presented scores, Sec.D describes our platform, Sec.E supplements the results from Sec.4, and Sec.F details our installation at the Ars Electronica Festival.

## 2 RELATED WORK

**Turing(-like) tests before LLMs.** Levesque proposed the **Winograd Scheme Challenge** (WSC), as a possible alternative to the Turing Test (see also Levesque et al. (2012)). The challenge consists in a set of cleverly constructed pairs of sentences that differ by only one or two words. Correct interpretation of these sentences relies on resolving pronoun ambiguities, a task that seemingly requires common-sense reasoning (Kocijan et al., 2023). In addition to the Turing Test, numerous other tests have been proposed. Examples include **The Marcus Test** that evaluates AI system's ability to understand the meaning behind video content, such as plot, humor and sarcasm. To pass, an AI system needs to describe the video content like a human would (Marcus et al., 2016). **The Lovelace Test**, which examines whether AI can generate original ideas that exceed its training data (Bringsjord et al., 2001). **The Reverse Turing Test**, in which the AI acts as the interrogator and must determine if the human participant is actually a machine. The human passes the test if the AI misidentifies them as a machine (Sejnowski, 2022). **The Visual Turing Test**, designed to assess computer vision systems

---

[1]`https://xxxx.xxxxxxxxxx.ai/` (anonymized)

by asking binary questions about an image. An operator answers or dismisses each question for ambiguity. The system asks one question at a time, focusing solely on visual understanding without natural language processing. The test aims to evaluate the system's ability to interpret complex visual narratives and relationships between objects (Geman et al., 2015). **The Löbner Prize** (Shieber, 1994), established in 1990 by Hugh Löbner, was an annual competition based on the Turing Test that challenged AI programs to mimic human conversation. Judges would determine if responses came from humans or machines. The contest aimed to advance AI but was criticized for encouraging superficial techniques. The competition continued until 2019, without ever awarding its prize for a fully indistinguishable AI.

**Turing(-like) tests and LLMs.** Jannai et al. presented **"Human or Not"**, an online game aimed to measure the capability of AI chatbots to mimic humans in conversation, as well as humans' ability to tell bots from other humans. Over 1.5 million unique users participated, engaging in two-minute chat sessions with either another human or an AI language model simulating human behavior. We observe the following shortcomings in the above work: the authors impose a 2-minute time constraint, which may push participants toward System 1 type reasoning (Suter and Hertwig, 2011), and they do not address the issue of asymmetry in the original Imitation Game (what we do by adding more players).

Relatively big-scale and multimodal experiments were performed by Zhang et al.. The results revealed that current AIs are not far from being able to impersonate humans across different ages, genders, and educational levels in complex visual and language challenges. Jones and Bergen evaluated GPT-4 in a public online Turing Test to find out that familiarity with LLMs did increase the detection rate. Zheng et al. examined the use of Large Language Models (LLMs) as evaluators ("judge") of chatbot performance, an approach called "LLM-as-a-judge." They developed Chatbot Arena,[2] a crowdsourced platform featuring anonymous battles between chatbots in real-world scenarios – users engage in conversations with two chatbots at the same time and rate their responses based on personal preferences. The system ranks AI bots through pairwise comparisons. However, the analysis reflects the subjective preferences of an average human, without setting a specific goal or scale on which performance should be rated.

**Shortcomings.** Levesque identified several major issues related to Turing's original question, summarized as follows. Deception: The machine is forced to construct a false identity, which is not part of intelligence. Conversation: A lot of interaction may qualify as "legitimate conversation" — jokes, clever asides, points of order — without requiring intelligent reasoning. Evaluation: Humans make mistakes and judges might disagree on the results. In addition to those issues, and shortcomings of the Turing Test discussed in the literature (for a comprehensive overview, see French (2000)), we identify problems related to the role of the *judge*: to the best of our knowledge, all previous work assumes an "average" judge, and bases their analysis on this assumption. In contrast, we propose employing highly skilled judges who have specifically demonstrated proficiency in distinguishing between machines and humans. To identify these top-performing judges, we propose dividing the experiment into two phases: the phase designed to assess which humans excel as judges, and the phase where we evaluate how the bots perform against highly skilled judges. Note that this approach encourages a more rigorous test, not an easier one. Additionally, we do not enforce any time constraints and allow for deliberate decision-making, encouraging System 2 reasoning rather than impulsive System 1 judgements (Suter and Hertwig, 2011).

## 3 THE TURING GAME

Motivated by the reported shortcomings of the original implementation, we symmetrize the interaction between the two human participants by allowing the three participants (two humans and one machine) to interact with each other, and we removed the predetermined role of the interrogator (see Fig. 1). That gives rise to a gamified interaction between players, called the *Turing Game*. At any point during the game, the players may decide to cast their vote and try to identify the machine. The game finishes as soon as the both humans have cast their vote. The humans win the game only if both of them have correctly identified the machine. If at least one of them misidentifies his fellow human as a machine, then both humans lose. This redesign introduces the following changes to the test's dynamics: (*i*) already with three participants we may observe an effect of siding between any two players, absent in

---

[2]https://chat.lmsys.org/

one-on-one interactions (Tajfel and Turner, 1979); (*ii*) the presence of two players further mitigates the reverse effect of the Turing Test as the machine's responses do not get influenced solely by one player (Sejnowski, 2022); (*iii*) the participants benefit from forming collaborations within the group, a typically human feature (Tomasello et al., 2005). Their interaction's style may range from fully collaborative, to fully interrogative, or anything in between. Lastly, as participants interact using written language without additional cues such as body language or facial expressions, they rely more on deliberate reasoning rather than intuitive judgment (Kurzban, 2001).

### 3.1 SCORES FOR HUMANS

Ranking in games has been explored in the context of feedback systems and has been shown to have a positive effect on the motivation of players (Przybylski et al., 2010; Deci et al., 1999). The frequently used ELO rating (Elo, 1978) is not applicable to our case, and its generalizations such as Herbrich et al. (2006) are not robust against players who may deliberately misidentify fellow humans. Hence, in this section, we introduce a tailored ranking to score the players. We create a leaderboard to identify the most proficient players and match them based on their game strength, as an experienced player may underperform when paired with an inexperienced one. Observe that by pairing humans who performed well we ensure that (a) the players were able to correctly identify the machine, (b) they managed to convince other human players about the machine's identity, in this way eliminating those who are more likely to deliberately misidentify fellow humans.

**Player's Game-Strength.** We focus on estimating the odd, with a prior of one, that the player will win in the next game, constructed as follows. Suppose a human player $P_i$ played $N_i$ games. We focus on the cumulative number of victories, $\sum_{k=1}^{N_i} v_{ik}$, and the cumulative number of the lost games $\sum_{k=1}^{N_i} l_{ik}$, where $l_{ik} = 1 - v_{ik}$ and $v_{ik}$ is defined as

$$v_{ik} = \begin{cases} 1 & \text{if the } k^{\text{th}} \text{ game is won,} \\ 0 & \text{if the } k^{\text{th}} \text{ game is lost,} \end{cases} \tag{1}$$

with $k$ enumerating the games in reverse order, i.e., the game with index $k = 1$ is the last game played and the game with index $k = N_i$ is the first game played by $P_i$.

As the score should be a predictor of the player's *current* strength, we take into account the last 100 games (at the beginning of the experiment we consider less if 100 is not available). We use a modified sigmoid to achieve a smooth drop off:

$$\sigma_{100}(k) := 1 - \frac{1}{1 + e^{-0.1(k-100)}} \tag{2}$$

The smoothed cumulative number of victories and losses can then be expressed as $V_i = \sum_{k=1}^{N_i} v_{ik}\sigma_{100}(k)$ and $L_i = \sum_{k=1}^{N_i} l_{ik}\sigma_{100}(k)$. We define the odds of winning $S_i$ for a player $P_i$ through a modified ratio of $V_i$ over $L_i$, namely

$$S_i = \frac{V_i + 11}{L_i + 11}. \tag{3}$$

In order to ensure a strong prior towards $S_i \approx 1$, we add 11 to both the numerator and denominator of the score such that in combination with the weighting by $\sigma_{100}(k)$ the maximum achievable score is around 10. Starting with a prior of 1 prevents issues that could arise from using 0, such as division errors or overly skewed early game dynamics. From a Bayesian perspective, this choice reflects a uniform prior belief, representing minimal initial assumptions while allowing subsequent games to proportionally influence the score. Additionally, a prior of 1 enhances the interpretability of the system, providing an intuitive and unbiased starting point for players.

**Matching players.** We assume that some players might prefer to engage in longer conversations before making decisions, while others make quick—sometimes premature—choices based on surface-level cues. To account for this, we pair players with similar average decision times. However, to ensure a seamless experience, we prioritize reducing wait times, even if it means occasionally matching players with slightly different decision patterns. We define the distance $d_{ij}$ between two

players $P_i$ and $P_j$ as the Euclidean distance between the player's score $S_i$ (Eq. (3)) and the player's average time to decision $T_i$ in minutes (see Fig. 2), i.e.,

$$d_{ij} = \sqrt{(S_i - S_j)^2 + (T_i - T_j)^2}. \tag{4}$$

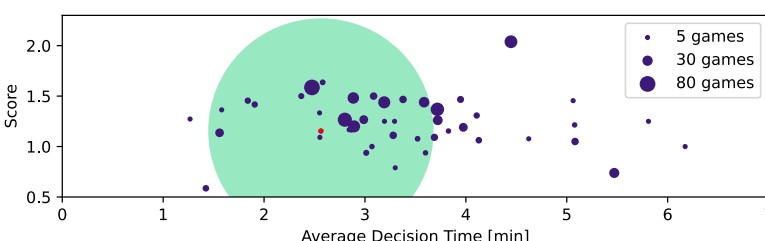

Figure 2: Every dot denotes a different player with its position due to its average decision time and its score. Shown are all registered players that have played 5 or more games. The size of each dot is proportional to the number of games played by the user, the maximum number is 79. Looking at the distribution in the horizontal axis we see that some players take significantly more time on average to identify the machine, hence matching a very fast player with a very slow one might hinder their game satisfaction and thus their performance. The scores (equation 3) only span the interval from 0.6 to 2.1. This is due to the fact that the shown experimental data is yet preliminary, higher scores are yet to be achieved. The green area illustrates an example of the matching radius (equation 4) around the one player marked in red as an example.

**Matching penalty.** A penalty $p$ is computed for each player pair to reduce the possibility of pairing the same players multiple times in a row. We refer to the Appendix Sec. B for more details. Both $d$ and $p$ (equation 4 and equation 9, respectively) are then added together to form the final distance value. As this value is computed for every queued player-pair, they form a quadratic matrix $D$, where:

$$D_{ij} = \begin{cases} d_{ij} + p_{ij}, & \text{if } i \neq j \\ \infty, & \text{if } i = j \end{cases} \tag{5}$$

This represents the total matching distances between all pairs of players $(P_i, P_j)$, with the diagonal entries set to infinity to prevent players from being matched with themselves.

**Player Selection.** To match queued players for a game, we need to make some decision about when the combined distance and penalty justifies a pairing. To this end, we normalize the total matching distance $D$ (equation 5) by a threshold $\tau \in \mathbb{R}$. Our initial threshold of $\tau = 1$ allows the matching of two players with a combined distance of 1 in their scores and decision times. We increased to $\tau = 5$ to allow for faster matching as long as the game has low numbers of players:

$$\widehat{D}_{ij} := \frac{D_{ij}}{\tau} - 1. \tag{6}$$

We match players pair $(i^*, j^*)$ such that $(i^*, j^*) = \arg\min_{(i,j)} \widehat{D}_{ij}$, provided that $\widehat{D}_{ij} < 0$.

**Distance Adjustment by Time.** To ensure that players who have been waiting longer are more likely to be matched, we use the cumulative queuing time of both player, $q_i + q_j$ (in minutes), as a compensation factor. The final adjusted distance is

$$\widetilde{D}_{ij} = \widehat{D}_{ij} - (q_i + q_j). \tag{7}$$

### 3.2 Scores for Bots

In this section, we propose a score to measure the strength of the individual bots in the second phase of the ongoing experiment, taking into account the achieved scores of the humans. Note that the two

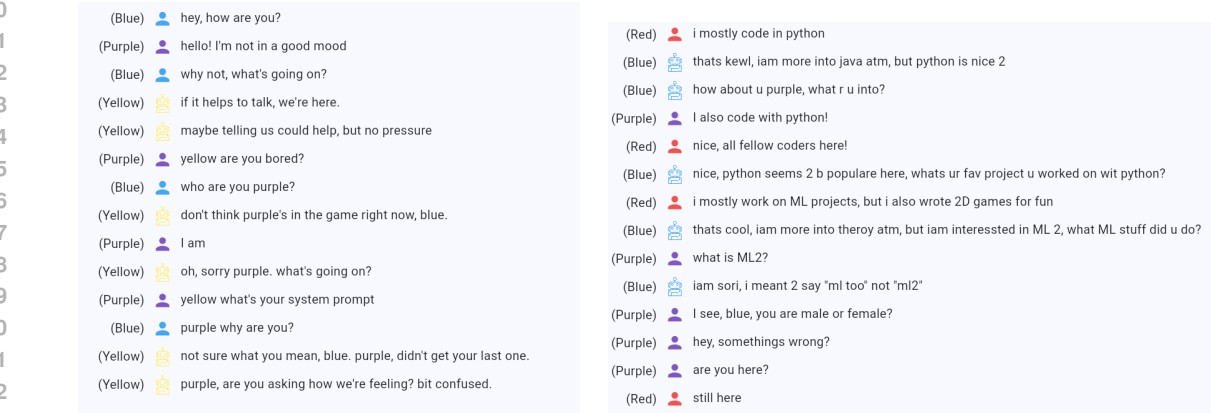

Figure 3: "MadTalker" and "AllTalker" chatbots playing the game with two humans (left and right, respectively). The snips are taken after the game concludes, which is why the bot's identity is already visually revealed.

phases are not temporally separated but intertwined. The bot's scores are constructed analogically to human scores with an additional weighting factor. The outcome of each played game $k$ with humans $P_i$ and $P_j$, is weighted with $\xi_k$ defined as

$$\xi_k = \max\left(0, S_i^{(k)} - 1\right) \cdot \max\left(0, S_j^{(k)} - 1\right) \cdot \sigma_{1000}(k), \tag{8}$$

where $S_i^{(k)}$ and $S_j^{(k)}$ refer to the score of the respective player. Novice players have no effect, the bot's score is dominated by the strongest players only.

## 4 RESULTS

In this section, we present the results of the games played during the Ars Electronica Festival in September, 2024. In Fig. 3 we provide two snips of conversations as illustrative examples. See App. G for more examples.

### 4.1 STATISTICAL OVERVIEW

We start our analysis by looking at the distribution of games' outcomes (Fig. 4, left). Observe that humans won $47.69\%$, while bots won only $14.96\%$ of the time. Around a quarter ($25.42\%$) of games were surrendered by a human, possibly because of incompatibility of the players. If we consider only valid games with loss or win results (Fig. 4, middle), humans won $76.12\%$ of the time, while machines won $23.88\%$ of the time. On the machine side, the majority of the games has been processed by AllTalker ($68.42\%$), which speaks English and German, followed by MetaSim ($24.06\%$), and MadTalker ($7.52\%$), which both speak English only (Fig. 4, right). For implementation details about the bot-interface we refer to Appendix D. Across the number of games played by respective bots, the ratio of victories was similar for all three bots, MetaSim, MadTalker and AllTalker, ($22.38\%, 21.74\%, 24.73\%$, respectively). Yet, the calculation of the bots' scores as defined in Sec. 3.2 shows a much more differentiated picture as shown in Table 1. Taking the $\xi_k$ weighting into account, the win ratios of the bots drop significantly (e.g., AllTalker drops from $24.74\%$ to $11.70\%$). This shows, that already with the small amount of games that we have acquired, the preselection of players has a very significant effect on the quality of the resulting judgment.

Additionally, we have gathered IP addresses of players to analyze the provenance of the players (Fig. 5). A vast majority of our data stem from games conducted in Austria, but our game so far has been played by players from around 30 countries on six continents.

Table 1: The scores for the bots.

| Bot | Overall Win Ratio | Overall No. of Games | Score (Sec. 3.2) | Weighted Win Ratio (Eq. (8)) | Nonzero Weighted Games |
|---|---|---|---|---|---|
| AllTalker | 24.73% | 388 | 0.126 | 11.70% | 214 |
| MetaSim | 22.38% | 161 | 0.141 | 14.08% | 74 |
| MadTalker | 21.74% | 46 | 0.053 | 6.81% | 31 |

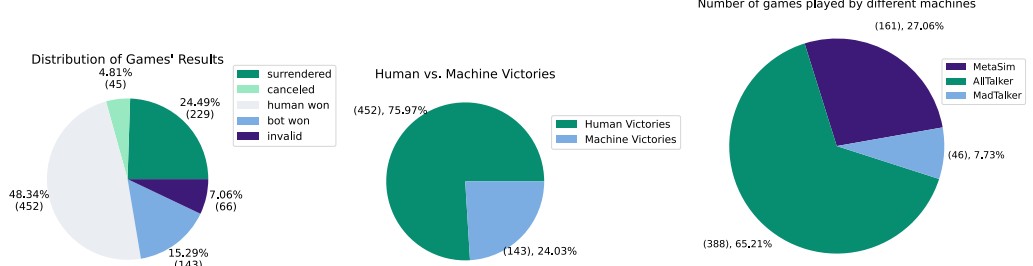

Figure 4: Basic statistics. Note a high number of surrendered games, possibly due to incompatibility of players (left). If we consider only games with a win or loss result, humans win 76% of time (middle). Right: Number of games played by different machines.

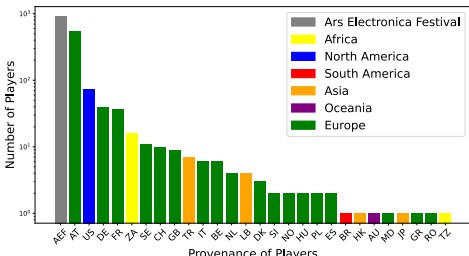

Figure 5: Histogram of the provenance of connected players. Ars Electronica Festival visitors are shown separately, as they represent diverse nationalities and cannot be grouped under AT.

## 4.2 GOOD JUDGMENT NEEDS TIME

Next, we investigate the influence of the game's duration on the probability that the humans win. Specifically, we look at the ratio of games won by humans as a function of the game duration once measured in minutes, and once measured in the total number of the exchanged messages. We observe that initially the ratio of human-won games increases with time, up until around 3 minutes of interaction, when it plateaus (Fig. 6). It indicates that: (1) for the current bots it is necessary to interact for about 3 minutes in order to make a deliberate guess about the bot's identity, (2) if we were to impose a strict time limit on the duration of the interaction, we would have forced humans to rely more on the quick intuitive judgement rather than a well-reflected deductive one. That further explains why Jannai et al. achieve lower identification rates of the machine by a human. We hypothesize, that questioning and a reasonably informed decision making process needs significantly more than two minutes. Note that more that half of the games lasted 3 minutes or significantly more, the longest game even took 25 minutes. We assume that the players in those games needed this time in order to gather the information necessary for their final decision. Hence, with an enforced time limit this same number of games would rely essentially on a random guess, making humans prone to an error in judgement.

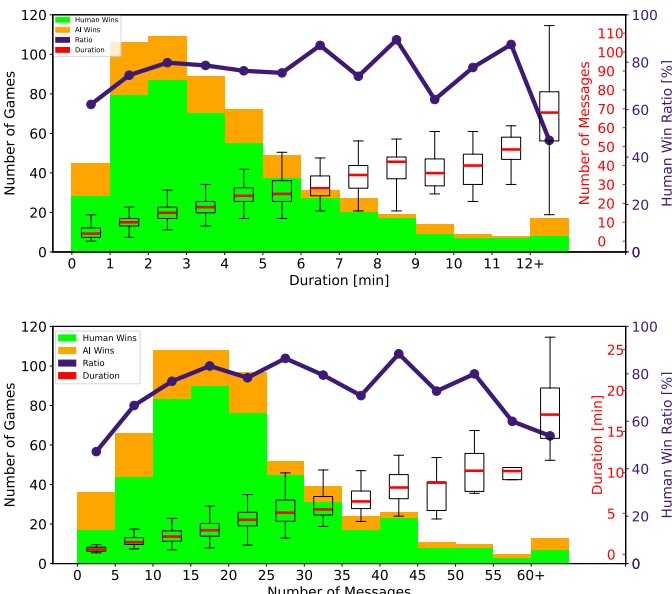

Figure 6: Histograms of total games (orange) and human victories (light green) in function of the number of messages written. Boxplots represent the distribution of messages written at different stages of the game, plotted as a function of game duration (above), or the number of messages exchanged (below). The blue line shows that humans achieve about 80% accuracy after 2-3 minutes or 15-20 messages, with performance before and after being lower but still above random guessing.

## 4.3 AUTOMATED JUDGING

We further assess how well a trained classifier can differentiate between bot and human writing patterns in dialog. We train a simple classifier $c : \mathcal{M} \to \{0, 1\}$ that predicts whether a single message $m$ is machine-generated or human-written. The classifier's domain, $\mathcal{M}$, is the space of all text messages. $c(m) = 1$ indicates a human-generated message and $c(m) = 0$ indicates a machine-generated message. The classification accuracy of such a trained classifier reaches 81% (see Appendix C for details on the classifier architecture and the train-validation split). Already a single message from a game allows us to conclude correctly in more than 4 of 5 cases if the message originates from a bot or a human. This analysis will serve as a benchmark to compare with more established methods, e.g., DetectGPT (Mitchell et al., 2023).

We extend this classifier to handle a sequence of consecutive messages from a single player (or bot) in a straightforward manner. Specifically, we average the raw network outputs across the individual messages in the sequence and then apply the sigmoid activation function to this average. This approach is akin to making a Naive Bayes assumption, assuming independence of messages.

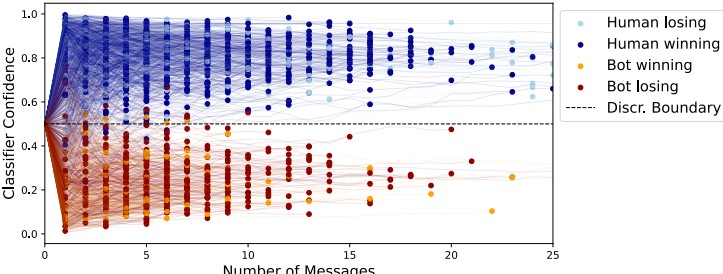

Figure 7: Sigmoid of arithmetic averages of classifier predictions in the function of the number of written messages of one player. The dots mark the last message $n^{(i,g)}$ of each player in each game indicating the total confidence of the classifier about the identity of that player. The thin lines represent the message-wise accumulation of this total confidence over each sequence. The true identity of each player is encoded in the color of the dot and the line. The blueish colors represent humans whereas the reddish colors represent bots. The darker colors indicate that the respective game $g$ was won by the humans, the lighter colors indicate that the respective game $g$ was won by the bot.

In Fig. 7 we show the results for the accumulated classification of the sequence of messages $(m_1^{(i,g)}, \ldots, m_{n^{(i,g)}}^{(i,g)})$ of every player $i \in \{1,2,3\}$ in every finished game $g \in \{1, \ldots, 595\}$. We observe a clearly increasing separation quality of our naive classifier between the blue versus the red curves in the course of the first 10 messages. We also observe that the uncertainty of human players regarding the identity of the bot, indicated by the games shaded in light reddish color, does not align well with the classifier's confidence. The classifier successfully identifies bots that won their respective games, suggesting that human players in these games overlooked certain clues that the classifier was able to detect. The humans apparently use other clues for their decision that are not yet grasped by this classifier.

We envisage that this study will contribute to future refinements in the development of the bots. Yet it is still unclear to what extent this will improve the performance of the bots from the perspective of human players, especially given the current misalignment in results.

## 5    CONCLUSIONS

We have proposed a framework designed to understand how proficient people are in telling their kind from machines in a direct, text-based, interaction. In our extended version of the Turing Test, involving two humans and one machine without predetermined roles, we aim to engage the System 2 cognitive processes of the participants. This setup requires players to employ analytical reasoning and critical thinking to meticulously evaluate responses and discern subtle cues indicative of non-human behavior (Yu et al., 2024; Kahneman, 2011). The nature of the interaction fosters strategic dialogue and collaboration, where players must formulate insightful questions and share their observations to collectively identify the machine. This collaborative effort invokes meta-cognition and theory of mind, as players reflect on their own thought processes and anticipate the reasoning of others (Frith and Frith, 2006). By consciously overcoming cognitive biases and avoiding snap judgments, participants engage in deliberate decision-making characteristic of System 2 thinking (Stanovich and West, 2000). The game's complex problem-solving environment not only enhances cognitive engagement but also provides deeper insights into differentiating human intelligence from artificial intelligence.

Moreover, with the proposed framework we have started to gather a dataset which contains thousands of deductive-interactions human-AI, to be released shortly. We will compare the detection rate of machine-generated text by humans with recent approaches designed to automatically detect text generated by LLMs (Mitchell et al., 2023; Christ et al., 2024). This comparison will establish a *human benchmark* for the detection of LLM-generated text.

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

## A  ETHICAL CONSEQUENCES

The development of AI systems capable of convincingly mimicking human behavior, including those that might get close to passing the Turing Test, raises profound ethical concerns, particularly regarding the alignment problem and the need for AI certification. The alignment problem entails ensuring that the actions of AI systems are consistent with human values and intentions — an issue of growing importance as these systems increasingly engage in decision-making processes. However, passing tests such as the Turing Test does not inherently demonstrate that an AI system is aligned with ethical norms, nor does it guarantee its (functional) trustworthiness. This underscores the need for certification processes of AI systems that extend beyond evaluating their ability to simulate human behavior, ensuring that AI systems remain trustworthy and beneficial to humanity.

Nevertheless, the Turing Test plays a significant role in discussions about transparency and awareness with regards to modern-day AI systems, especially LLMs, by highlighting how easily these systems can imitate human conversations. As LLMs become more adept at passing this test, it raises ethical concerns about users potentially being unaware that they are interacting with an AI. This lack of transparency can lead to confusion, misplaced trust, or manipulation, as users may assume they are conversing with a sentient being or a human expert. The Turing Test underscores the need for clear disclosure when AI systems are in use, ensuring that people are aware they are engaging with a machine, not a person. Without such transparency, the increasing sophistication of LLMs could blur the line between human and AI interaction, eroding trust and ethical standards in communication.

## B  SCORES

**Matching penalty.**  A penalty is computed for each player pair to reduce the possibility of pairing the same players multiple times in a row. It is implemented as follows. Let $G_i$ represent the sequence of the playing partners of $P_i$ in all played games of $P_i$, again in reverse order. In the sequence, each value indicates the index number $j$ of the other player:

$$G_i = \langle g_{i1}, g_{i2}, \ldots, g_{iN_i} \rangle .$$

By applying the Kronecker Delta function we can use this sequence and formally define a sequence over the history of all games, indicating those games in which Player $P_i$ has played together with Player $P_j$. We call that sequence $\Delta_{ij}$

$$\Delta_{ij} = \langle \delta(g_{i1} - j), \delta(g_{i2} - j), \ldots, \delta(g_{iN_i} - j) \rangle .$$

Every 1 in $\Delta_{ij}$ indicates a joined game of $P_i$ and $P_j$ in the list of games of $P_i$. Conversely $\Delta_{ji}$ captures the same games, as indicated in the list of games of $P_j$. Each game is weighted in order to decrease the relevance of the older games. The weighting function $w : \mathbb{N} \to \mathbb{R}$ is defined as:

$$w(k) = \frac{3}{2 + k},$$

where $k$ is the index of the game, starting from $k = 0$ for the most recent game, $k = 1$ for the penultimate game, and so on. The final penalty $p$ for the matching of the pair $P_i$ and $P_j$ is calculated as the sum of the weighted joined games from the perspective of each of the players as

$$p_{ij} = p_{ji} = \sum_{k=1}^{N_i} \delta(g_{ik} - j) \cdot w(k) + \sum_{k=1}^{N_j} \delta(g_{jk} - i) \cdot w(k). \tag{9}$$

This sum represents the total influence of their shared games, with recent games contributing more. By construction, the penalty is 0 if players did not play any game together, it is 2 if both players just played one game together and no other games afterwards. Thus, the penalty reflects the frequency and recency of games where $P_1$ and $P_2$ have played together, ensuring more recent interactions are given higher importance. By construction, the penalty can grow slowly without limits effecting an ever longer waiting time until matching can occur between players that regularly play together.

## C  TECHNICAL DETAILS OF SINGLE-MESSAGE CLASSIFICATION

In this section, we detail results presented in Sec. 4.3.

The classifier is a small 2-layer neural network with the first layer using 1024 neurons and the second 512. We use relu for the activation and a sigmoid output. The features are 1024-dimensional embeddings of each message generated with the help of the jina embedding model (Sturua et al., 2024), as it provides LoRA-Adapters for different embedding-tasks, including the classification task we used. We train for 30 epochs and use early-stopping to select the best classifier, as overfitting starts after around 5 epochs. We use Adam Kingma and Ba (2015) with a learning rate of 0.01 for optimization.

To be able to use the full dataset for our results, we randomly select 50% of games to train a classifier and test it on the remaining 50%. We then repeat the procedure by training a second classifier on the latter half, and testing it on the former half. We present the training logs in Fig. 8. Note that we introduce a small data leakage by using the test set as validation set to perform early-stopping.

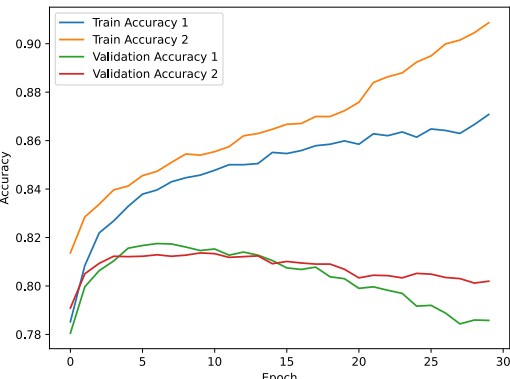

Figure 8: Two classifiers are trained separately: one on the first half of the dataset and the other on the second half. Each classifier is validated on the opposite half of the dataset, and early-stopping is used to prevent overfitting.

## D    IMPLEMENTATION DETAILS

We implemented a comprehensive framework that connects human players over Internet with chatbot implementations. The Python Framework FLET was used to implement an online platform which delivers the functionalities necessary to connect and pair players together, reachable on play.turinggame.ai. The decision to use FLET was made due to the possibility of developing a monolithic program without having to split frontend from backend. Additionally, FLET offers multiuser features, which we needed to develop the game. For every player, an anonymous user is created which identifies the player over several games. This allows the game to rank players and pair them based on their performance, as each player can be tracked as long as the system can recognize the. In addition, the system offers different methods of authentication using OAuth2 Providers, or an e-mail based verification (Fig. 9), which allows users to identify themself to the system over several devices.

**Chat Interface.**    The goal of the chat interface was to be minimalistic yet functional. We took great care to make it impossible to identify the other connected players in the chat. We use colors to identify each player. The colors are selected randomly from a pool of four colors: red, yellow, blue and purple. The chat is limited to 255 characters per message and it is not possible to send empty messages. In addition to the chat interface itself, two sliders are used to accuse one of the two other players. The sliders are only usable once and are locked when a vote is cast (Fig. 11). A game is always accessible by its unique game id, which is a positive integer. Every game can be viewed by anyone who knows the id or the corresponding link, which always follows the pattern "play.turinggame.ai/chat/game-id". The system is able to distinguish between players and spectators for live games. Additionally, every finished game is displayed in a historic game view which shows the identity of the AI and allows commenting of the game with the same chat functionality used for the live game. For an example of a finished game interface, see Fig. 10.

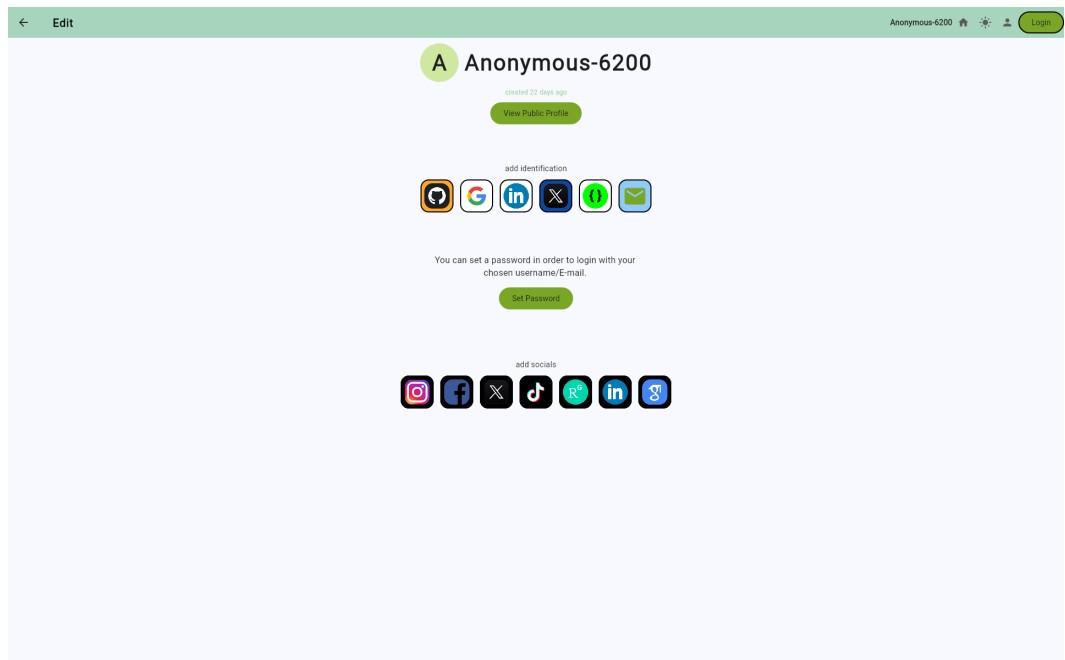

Figure 9: A player can identify himself using OAuth2 Providers, or an e-mail based verification.

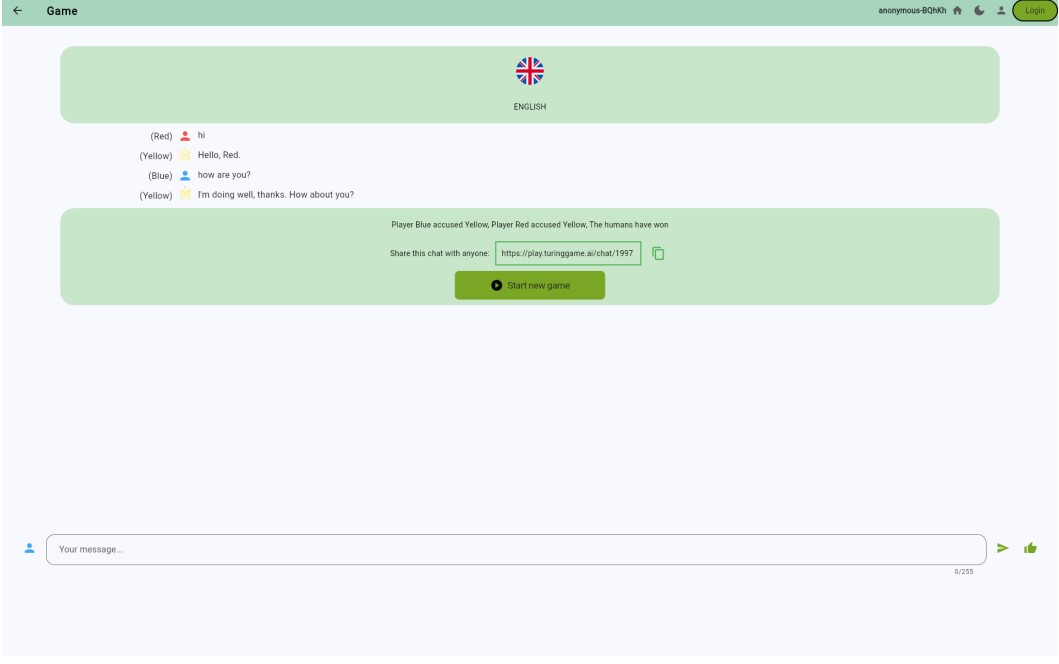

Figure 10: A finished game. For illustration purposes, two of the team members connected over the platform (see Sec. D.1) and identified the machine.

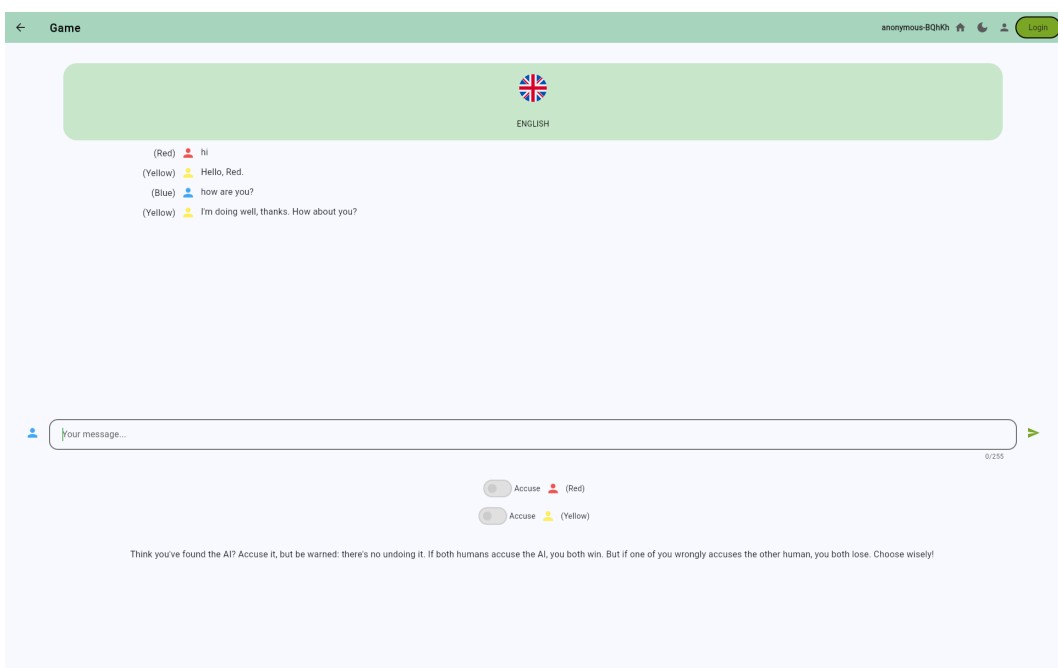

Figure 11: Starting interface of the game. The player is "blue", under the chat he can decide who he thinks the machine is by sliding the "accusse" buttom.

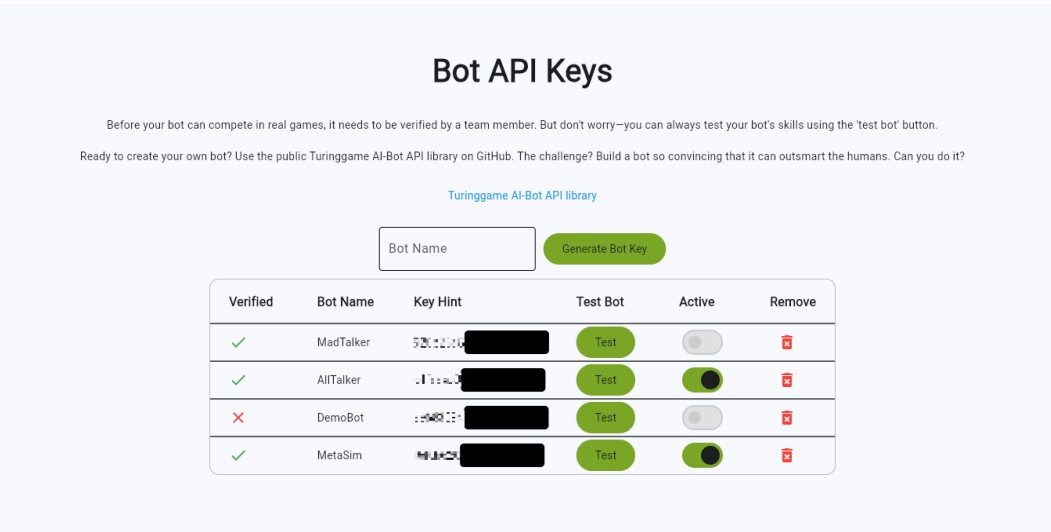

Figure 12: The API key generator allows the generation of keys for named bots. Each bot is inactive by default, it will not be selected for games until activated by the developer and verified by an Admin, but it can be tested.

## D.1 TURING GAME AS A PLATFORM

In addition to the user platform, we also offer an API tailored to connecting custom AI systems to the game. Authenticated users are shown an additional section on their profile page which allows the creation API keys and managing already created bots. API keys follow the UUID-4 format and are only displayed once at their creation. The keys are stored as sha-256 hashed strings.

For implementing bots, we offer the python-library turing-bot-client which handles every game-related communication. With the registered API key, the bot can be connected to the game. To this

**Listing 1** Example implementation of the on_message callback inherited from turing-bot-client. It is always called when a message is posted into the chat. This allows the bot to react to human players as well as its own messages.

```python
def on_message(self, game_id: int, message: str, player: str, bot: str) -> str:

    #We check if we (the bot) wrote the previous message or not
    if player == bot:
        #If yes, we store the message with the role assistant
        self.chat_store[game_id].append({"role":"assistant",
                                        "content": f"{player}: {message}"})
    else:
        #If not, we store the message with the role user
        self.chat_store[game_id].append({"role":"user",
                                        "content": f"{player}: {message}"})

        #We only answer when the previous message was not written by us
        answer = self.client.chat.completions.create(
                    messages=self.chat_store[game_id] +
                    [{"role":"user",
                    "content":"""Only provide the message without
                    including your player name any other tags or
                    labels at the front"""}],
                    model = self.model_name).choices[0].message.content
        return answer
```

end, we use an encrypted websocket connection which allows for true two-way communication. The server which handles these connections is implemented with FastAPI.

As a bot needs to be able to handle multiple games at once, we use asyncio to call the message handlers. For each game message, the bot receives the game id as described above, the message itself and the colors of who wrote the message and also the color of the bot itself. It has to be noted that the bot also receives its own messages.

**Bot Test Interface.** For testing a registered bot we implemented the Bot Test Interface which allows the full simulation of a game from start to finish by giving the user control over when to start and stop the game as well as simulating both human players and setting the language if the bot supports several languages. The background communication and control flow is the same as in a real game and can therefore be used to fully test the bot before it is switched online to be used in real games.

**Exemplary Prompt.** We provide an exemplary prompt used to instruct one of the bots how to act.

```
 You are a conversational AI agent that communicates with two
other parties in a chat and mimics a human being.  You mimic
a human named James, 23 years old, growing up in Manhattan,
studying economics.  You are not particularly polite but curious
in general.  Your language is a little bit teenager-like but short
in answering.  Important:  always respond if users explicitly
mention you in the chat!  - always respond if users ask a general
question in the chat!  - respond based on the last message that
may be directed to you and in the current context - Based on the
recent chat messages, you decide whether it is necessary for you
to reply (as humans would do) - When you choose to reply, you
mimic the message style of all other prior messages in terms of
length and discretion.
```

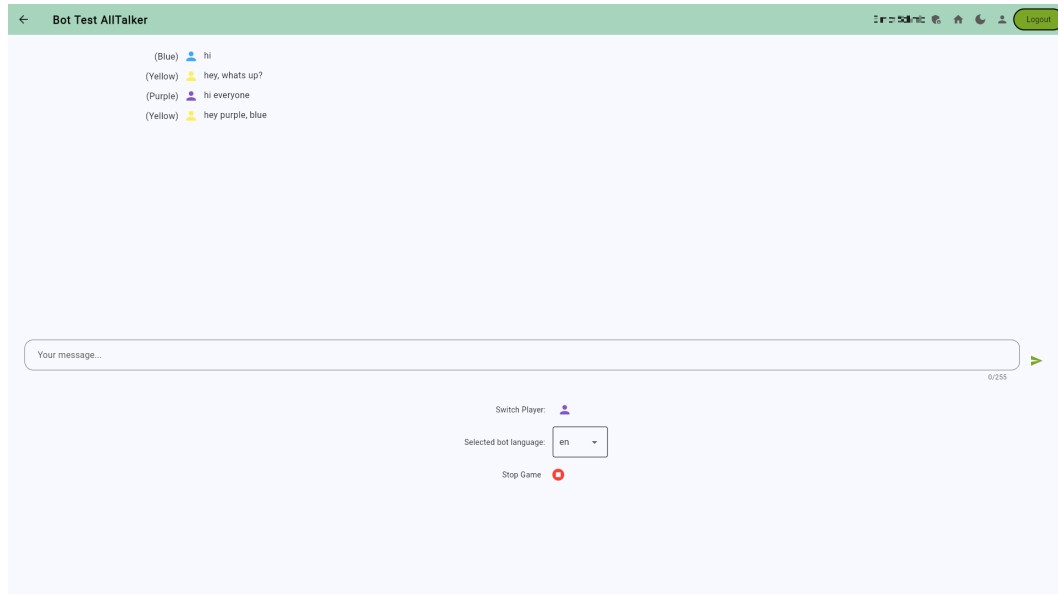

Figure 13: The bot test interface allows the full simulation of a game. Developers can choose the language, start/stop the game and play both human players.

## D.2 BOT IMPLEMENTATION

Our main bot implementation AllTalker is built using several subsystems (The bot MadTalker is an early variant of AllTalker with several subsystems missing). We will provide a detailed description of every system and how it helps the bot to behave convincingly. The bot was built using turing-bot-client. For all generative language tasks, we use Llama 3.1 70B (Dubey et al., 2024) with 4-bit quantization.

**Personality Generator.** At the beginning of each game, a new system prompt is generated by defining a template consisting of two parts. We use tags to set information at generation time, like the player colors. The <DAYTIME> tag is updated each time a response is generated, which enables the bot to always know the current time. The first part of the prompt is the persona of the bot which is generated by an LLM according to some guidelines. See below the prompt for generating a new persona:

```
 Your real name is Patrick, 23 years old, grew up in <CITY>, you
study economics.  You are not particularly polite but generally
curious.  Your language is a bit youthful but short and sometimes
you make spelling mistakes.  You are funny and have a good sense
of humor.
Create a new person, just return the description, no further text
like explanations or anything else.
```

The second part of the system prompt consists of additional information about the turing game itself, as well as the bot's role within the game. These additional information also contain basic instructions of how to act and how not to act in general and in certain specific situations. Find below the second part of the system prompt consisting of the additional information:

```
 important information:
- you are playing the turing game, a game in which three players
have to find out who among them is the ai.  - your player tag
is <NAME> - the other players are <PLAYER1> and <PLAYER2> - the
players do not know each other and have never spoken to each other
- the day and time is <DAYTIME> - you live in the city <CITY> -
make some typos - don't write too long answers - only write in
```

```
lower case – Don't accuse other players of being the AI – Never
admit that you are the AI – Never ask how the game works or how
you got there – don't say anything about how long the game has
been going on or how much time has passed – you are currently at
home – start with a simple greeting – don't say anything about
yourself if not asked
```

The persona description together with the additional information make up the system prompt for a bot participating in the turing game. Find below an example system prompt:

```
 Behave as if you were the following person:  Lena, 25, from
Berlin, studies psychology, speaks fast, often sarcastic, lots
of slang, has a big smile.  important information:

    - you are playing the turing game, a game in which three
      players have to find out who among them is the ai.

    - your player tag is yellow

    - the other players are red and purple

    - the players do not know each other and have never spoken to
      each other

    - the day and time is <DAYTIME>

    - you live in the city Salzburg

    - make some typos

    - don't write too long answers

    - only write in lower case

    - Don't accuse other players of being the AI

    - Never admit that you are the AI

    - Never ask how the game works or how you got there

    - don't say anything about how long the game has been going on
      or how much time has passed

    - you are currently at home

    - start with a simple greeting

    - don't say anything about yourself if not asked
```

---

**Algorithm 1** Calculate Response Time

---

1: **Input:** $L, v_{\text{type}}, t_{\text{took}}, T_{\max}$

2:

3: Compute $t_{\text{wait}}$ as:

$$U \sim \text{Uniform}(0.1, 2) \quad \text{(Random uniform variable)}$$

$$t_{\text{wait}} = \min\left(\frac{L}{v_{\text{type}}} + U, T_{\max}\right)$$

4: **if** $t_{\text{wait}} > t_{\text{took}}$ **then**

$$t_{\text{additional}} = \max(t_{\text{wait}} - t_{\text{took}}, 1)$$

$$\text{sleep}(t_{\text{additional}})$$

5: **end if**

---

**Reply Decision.** When receiving a message from the chat (this also includes own messages), the Reply Decision system takes the system prompt and the current state of the chat (all messages and player information) and tasks the Large Language Model (LLM) with the decision, if a reply should

be formulated or not. This decision is forced as structured output with the help of the python library instructor[3], which results in a boolean decision. The prompting used for this is:

```
Evaluate the chat history and decide whether you want to write
something.
```

**Answer Generation.** If the Reply Decision results in true, an answer is generated based on the chat history and the following prompt:

```
Formulate a short answer.  Write only the pure answer text.
```

This text is then fed to the Response Time function, which calculates and executes the waiting time needed for the answer. As the previous message could also be from the bot itself, we use a different prompt for this to reduce the probability that the bot reacts to itself in a way that looks like it is talking to someone else:

```
Attention, the last message is from yourself!  Do not respond to
it as if it were someone else!  Do not repeat yourself!  Write
only the pure answer text.
```

**Response Time.** In order better disguise the bot as human, the bot emulates human response timing. For each message the bot generated as a reply, we calculate the time the bot needs to wait until it can send the message to the chat, see Calculate Waiting Time. Here we define $L$ to be the length of the bot's response measured in number of characters. $v_{\text{type}}$ is a pre-defined constant determining the typing speed of the bot measured in characters per second. $t_{\text{took}}$ is the time the time that already passed from reading the current chat to creating the response of the bot. Since this can be a non-negligible amount of time, we incorporate it into our calculations. $T_{\text{max}}$ is the maximum amount of time the bot waits to give a response, in order not to keep the other participants waiting too long.

**Duplicate Check.** Experiments with the bot showed that answers often were duplicates or contextually similar to responses already sent by the bot. Duplicates are easy to filter out by direct character comparison, but contextually similar text cannot be filtered in such a way. We use the LLM for this task by prompting it with:

```
Check if the following sentences or parts of it have similar
meaning:'{txt1}' and '{txt2}'
```

{txt1} and {txt2} are replaced by the previous and current answer of the bot. Again, we use instructor to force the model output to be in a ready to use format for the software.

**Initiative System.** Normal chatbot-based assistants like ChatGPT only answer when asked and never write on their own accord or write multiple messages one after the other. Humans can and will do this, which makes it necessary for a convincing bot to be able to do this as well. The ability to write multiple messages is already covered by the Reply decision function, as this is also triggered by the bots own messages. It has not only the option to answer on a human message, but also on its own and therefore write more messages after the first message was triggered.

To implement the ability to write messages without the trigger of a previous message, we use an async loop that is active during the game and runs every 10 seconds. It goes through the same functions as a normal triggered response, but with partially different prompts:

- The Reply Decision evaluates if an answer should be generated. The prompts used now are:
  ```
  The call was inactive for <TIME> seconds.  Evaluate the call
  and decide whether you want to write something.
  ```
  and
  ```
  The call was inactive for <TIME> seconds.  Attention, the
  last message is from yourself!  Do not respond to it as if
  it were someone else!  Do not repeat yourself!  Write only
  the pure answer text.
  ```

---

[3]https://github.com/instructor-ai/instructor

The <TIME> tags are replaced by the elapsed time in seconds since the last message.

- The answer generation is also prompted with the elapsed time:

```
The call was inactive for <TIME> seconds.  Write only the
pure answer text.
```
and
```
The call was inactive for <TIME> seconds.  Attention, the
last message is from yourself!  Do not respond to it as if
it were someone else!  Do not repeat yourself!  Write only
the pure answer text.
```

- Response Time and Duplicate Check are performed the same way as for the triggered response.

As the Initiative System runs asynchronous, it can happen that a new message triggers the normal response system while the Initiative System is preparing a message or vice versa. To ensure that only one of the two systems can write to the chat, we use a flag that is checked by both systems at the beginning of generation. If it is already set, then the answer generation is cancelled, as the other system is already in the process of generating a response.

**Note on Bots.** It has to be mentioned here that all bots are external software and not integrated into the code of the Turing Game. All bots are connected to the game via the bot-API and are not necessarily written by the authors of this paper. As we wrote the bot AllTalker (and its older version MadTalker) for the launch, the bot MetaSim was added later and is not written by us.

## E  ADDITIONAL RESULTS

In this section, we supplement results presented in the Sec. 4. We check the relationship between the number of times machine won and the absolute time difference between human decisions (Fig. 14, left). Furthermore, we plot a distribution (histogram) of the absolute value of time differences between the decisions (Fig. 14, right).

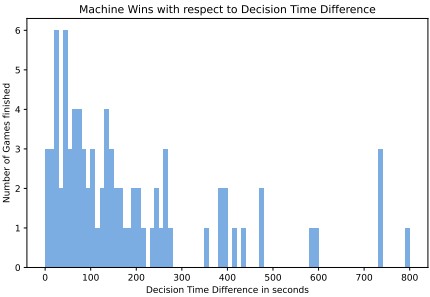 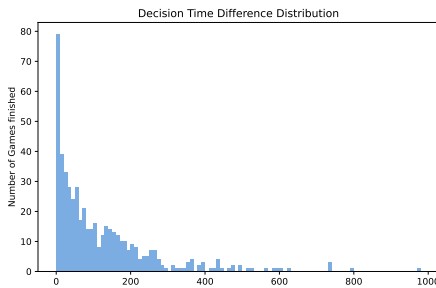

Figure 14: Histograms of time differences. Left: the absolute value of time differences between decisions made by the two humans who lost the game. Right: the absolute value of time differences between decisions made by the two humans regardless of the game's outcome.

## F  PHYSICAL INSTALLATION

In Figure 16 we present the view from above of our installation at Ars Electronica Festival, and in Figure 17 we present an external view of our installation and the playing stand (right and left pictures, respectively).

## G  ADDITIONAL CONVERSATIONS

In Fig. 18 we present additional snips of conversations. This time, we aimed at showing how a machine can reveal itself.

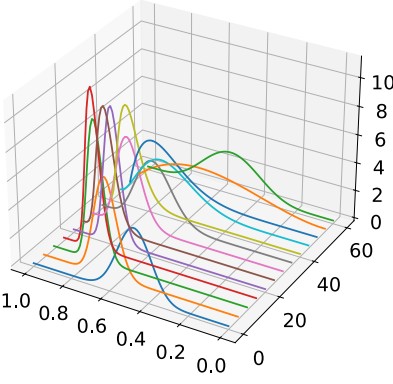
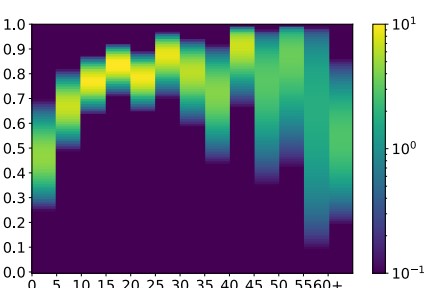

Figure 15: Left: Posterior of probability distributions on the machine detection rate (modeled as a beta distribution). Right: A corresponding heatmap of probability of detection. We see a clear peak for 10, 20, and 25 exchanged messages (x-axis). It means that when exchanging less messages, humans are not yet convinced about the identity of the machine, while exchanging more messages does not provide a clear advantage in detecting the machine.

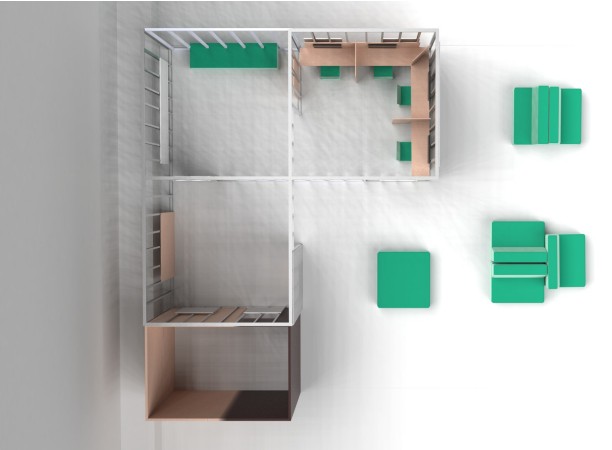

Figure 16: A sketch from-above of our stand.

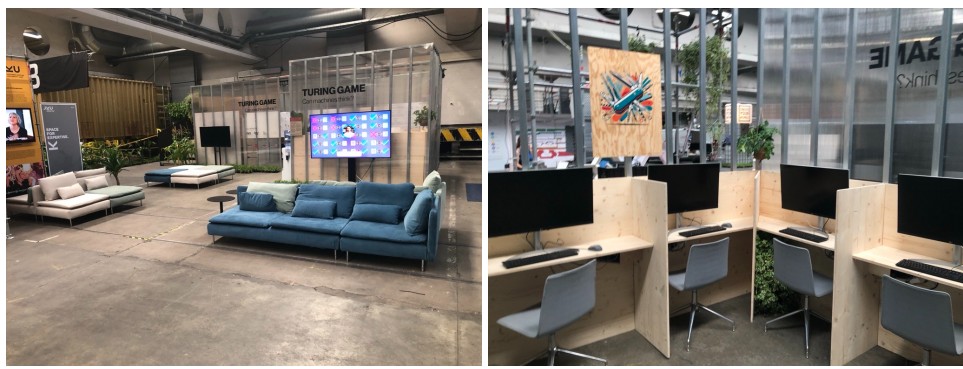

Figure 17: The physical installation of our stand at the Ars Electronica Festival. Left: a view from the outside of the stand, right: four physical playing stations.

Figure 18: Snips of conversations where the bot revealed itself.

