# OpenReview forum: "The Turing Game"
_ICLR.cc/2025/Conference — ICLR 2025 Conference Withdrawn Submission_

### Official Review · Reviewer_2FRu · 2024-11-01

**Soundness:** 2
**Presentation:** 2
**Contribution:** 2
**Rating:** 3
**Confidence:** 3

**Summary:**

The authors introduce a variant of the Turing Test: the Turing Game. The generalized Turing Game removes the interrogator role and focuses instead on the interaction between an AI and two humans. In addition, the authors walk through a deployment of the Turing Game: they propose ranking metrics and matchmaking methods.

The authors deploy their game and recruit human participants. Then they report statistics of their dataset; an analysis of length and human success; and the design and evaluation of a classifier that predicts outcomes in the Turing Game.

**Strengths:**

The task setup and dataset are very promising. Turning the game into a collaborative “who is the mole” game sounds useful. I’m additionally impressed by the infrastructure and effort that likely went into deploying this system—the deployment at Ars likely took quite a bit of work!

**Weaknesses:**

I worry that the infrastructure is too large of a focus of the paper. While the infrastructure and pairing methods are undoubtedly important, I found myself more interested in the dynamics of human-AI interaction as the game progresses. While the authors conducted an automated eval and a length-based analysis of success, there wasn’t a deeper analysis of _why_ these outcomes occurred or were observed.

In addition, I worry that some of the metrics are overcomplicated and have some arbitrarily chosen hyperparameters. Why derive your own ranking metric for this game when other ELO generalizations exist? See TrueSkill [0]. I think it satisfies many of your constraints. The bot itself could be treated as a single player in the TrueSkill setup, in it’s own team.

Finally, I worry that portions of the paper are incomplete. I’m not sure what the MadTalker, AllTalker and MetaTalker systems are. Given all these considerations, I worry that the authors do not have enough time to revise the paper in the rebuttal period.

[0] Herbrich et al. TrueSkill™: A Bayesian Skill Rating System, NeurIPS 2007.

**Questions:**

**General Questions:**

What are AllTalker, MadSim, and MetaTalker? I recommend introducing these methods before discussing them in the results.

There are some typos/spacing issues in the paper that a closer look might be able to help. For example, the grammar for the caption in FIgure 4: “he snips where takes once the game finished, that’s why the bot’s identity is already visually revealed.” could use some rewording. Also “nominator” -> numerator.

Why not just use an already-validated player ranking metric like TrueSkill?

Finally, did authors take consent from participants or pay participants? Was the user study approved by some kind of IRB?

**Framing Questions:**

I think the authors have a goldmine of a dataset in their possession. Unfortunately, the current paper’s framing focuses a bit too much on the system's infrastructure rather than the game's dynamics! The dynamics here are the interesting part: how do humans build common ground with each other in unsuccessful games? What about successful games? What cues do people rely on to single out an AI model? If longer interactions are indeed more successful, can we forecast early if a conversation will go long? Are early interactions likely to predict if users will successfully identify the AI?

More generally, I wanted to see a deeper analysis of the dynamics of how human successfully came to discern that the system was or wasn’t an AI. Also: why not reframe the paper and focus on this being a dataset contribution? That would be cool too. Are the authors planning on releasing all the interaction data?

**Details Of Ethics Concerns:**

Not sure how consent was taken, or if there was an IRB protocol process. Were participants paid? Unclear.

---

> ### Comment · Reviewer_2FRu · 2024-11-25
> **Thanks!**
>
> Read your rebuttal, along with concerns from the other reviewers!
>
> Unfortunately, I am keeping my score. I think players deliberately misidentifying is an odd case; doesn't that just make them a bad player? Guessing randomly, however, is considered in TrueSkill (like it is with ELO). The assumption here is that, in expectation, guessing randomly means you're a bad player (low ELO low trueskill).
>
> The last bit re: framing-
>
> Now those observations you were talking about are SUPER cool (re: naive / experts). I think you should honestly reframe around those analyses, use TrueSkill, and position your work as a dataset paper for a later submission. I also think there are some interesting ToM references / papers you can connect to too!

---

### Official Review · Reviewer_Dbbj · 2024-11-02

**Soundness:** 2
**Presentation:** 2
**Contribution:** 2
**Rating:** 3
**Confidence:** 4

**Summary:**

This paper introduces the Turing Game, a modern adaptation of Alan Turing's original one designed to test whether humans can distinguish between fellow humans and AI in chat-based interactions. The game features two human players and an AI chatbot, all interacting anonymously through text. Unlike the traditional Turing Test, which focuses on a one-on-one interaction with an interrogator, the Turing Game is symmetric, removing predetermined roles and encouraging collaborative or interrogative dynamics among the participants. The authors have launched the game on the Internet and gained preliminary results involving many human subjects.

**Strengths:**

I really like the idea to changing the Turing test into a game – humans in their nature want to win – this gives them the initial and the biggest motivation to take it seriously. This could avoid two challenges in vanilla design: (1) human subjects being perfunctory so that human judges can easily distinguish, and (2) human judges being perfunctory and guess randomly, invalidating results, since if they do so, they cannot win the game.

+ New design on Turing test.
+ Detailed metrics for matching players.
+ Large scale experiments on humans with deployment on the Internet.

**Weaknesses:**

- Research questions are unclear.
- Lack experiments for deeper analysis. The paper selects MadTalker, MetaSim and AllTalker without introducing them. We have no idea how the methods are implemented and how to interpret the results on these models. Instead, ICLR readers might be more interested on how specific models, say GPT-4o, LLaMA-3.2 perform.
- Paper presentation needs further improvement. It looks like a technical report instead of an academic paper. Also, many parts in the current version of the paper seem unnecessary for ICLR readers (e.g., the background of Turing test, the definition of Euler distance).

**Questions:**

1.	About the player rating: current metric considers only the victories and lost games. But I think some lose might be due to another human making a wrong decision. Current metric just counts it as a lost game – which is unfair for the human making the correct decision.
2.	How do you handle turn taking? This is not introduced in the framework session. Do you let the three parties speak one-by-one (roundtable style)? Or do you let them speak freely without restrictions? Then how does a chatbot decide when to stop taking inputs and generate responses?
3.	In the human collaboration design. Will there be bias when one human participant is more dominant?
4.	About the top-performing judges: do you have a threshold to determine whether human players are top-performing ones? What is LLMs’ performance (win rate) when the two human players are both top-performing ones?
5.	In ethics statements, I think you should discuss how to protect participants’ privacy, especially their demographic information and their chat history.

Presentation (minor) issues:

1.	Upon seeing figure 1 and 2, one cannot tell whether this game is about human (the male) competing against another human (the female), or two humans competing against a bot.
2.	Quotation marks need to be consistent (curly quotation marks recommended). There are still straight quotation marks at line 134, 145, etc.
3.	The related work has only been introduced but not been sufficiently compared with the proposed method. Need more comparison to show the difference.
4.	Table 1: There should be a period at the end of the caption. The header will look better if the first letters of words are capitalized.
5.	Figure 5: sizes of subfigures are inconsistent.
6.	Overall, the writing should be improved.

---

### Official Review · Reviewer_Zs5n · 2024-11-02

**Soundness:** 3
**Presentation:** 2
**Contribution:** 2
**Rating:** 6
**Confidence:** 4

**Summary:**

This paper introduces a novel approach to the Turing Test called the Turing Game, which symmetrizes the roles of human participants and removes the interrogator role present in previous implementations. The goal of the Turing Game is to assess whether machines can convincingly mimic human behavior in chat-based interactions, focusing on collaborative dialogue and intention inference as key elements of human communication.

**Strengths:**

The Turing Game brings a fresh perspective to the Turing Test by transforming it into a three-player game with symmetrized roles, where both humans participate in identifying the AI.

The creation and public launch of the Turing Game platform is a significant contribution to AI research.

The player-matching algorithm pairs participants with similar skills and decision-making tempos, ensuring that users have a more engaging and balanced game experience. This is a thoughtful design choice, as it minimizes frustration for players of different skill levels, leading to more consistent, high-quality data.

**Weaknesses:**

The paper provides a statistical analysis of game outcomes, but it lacks a deeper qualitative analysis of the conversations themselves. Examining specific examples of successful and unsuccessful bot interactions, and analyzing the types of conversational strategies employed by humans and bots, would provide richer insights into the dynamics of the Turing Game.

The preliminary data is drawn mainly from interactions at the Ars Electronica Festival, which may not represent a diverse user base. Participants at the festival may share certain cultural or technological backgrounds that could affect how they perceive and interact with AI.

**Questions:**

What are the specific details of the three bots evaluated in the study (AllTalker, MetaSim, MadTalker)? More information on their architecture, and training data would help readers understand their strengths and limitations.

How do the authors ensure that human players are sufficiently motivated to engage in the Turing Game and provide accurate judgments? What measures are taken to prevent players from deliberately misidentifying fellow humans or guessing randomly?

---

### Official Review · Reviewer_XJMd · 2024-11-05

**Soundness:** 2
**Presentation:** 2
**Contribution:** 2
**Rating:** 3
**Confidence:** 3

**Summary:**

This work describes the Turing Game, a modification of Turing's Imitation Game by setting up humans to take on a symmetrical role of conversing with one another in a three-way fashion to detect which one of the players is a machine, rather than the traditional setup of having one person take on the role of a  interrogator and another take on the role of a judge. This work also describes an elaborate setup for scoring and pairing players at similar skill levels.

**Strengths:**

- This work presents an interesting modification to the original Turing's Imitation Game by making adjustments in order to address shortcomings found in previous efforts of modern adaptations of this game.
- The paper presents in great detail the process and importance of scoring and matching players.
- The authors also present a classifier that can automatically identify the bot in the Turing Game correctly 81% of the time.

**Weaknesses:**

- While the design of the Turing Game is very interesting and the author describes the matching process in great detail, it lacks key information or appropriate explanations for interpreting the results (e.g., it is unclear even after reading the full paper, including the appendix what the different types of chatbots are, and how equation 8 is used to compute the adjusted win rates of the bots). Without knowing what the chatbots are and knowing their performance from previous adaptations of the Turing Test, the value of the Turing Game is unclear to me. For example, do we find that these bots are considered indistinguishable to humans in previous work while the Turing Game finds that they are clearly distinguishable, as demonstrated by the low win rate?
- In general, the work lacks clarity and requires many improvement in terms of presentation. See questions/suggestions for further detail.

**Questions:**

- Incorrect citation format. Differs from  other papers that I've reviewed.
- Figure 1 and lines 64-68 are not clear on what the redesign is and how this is a better setup than the original Imitation Game. It would help to elaborate on what the checks and crosses mean, along with the hammers and the arrows.
- lines 69-71: How does this setup avoid the quicksand of the notion of thinking?
- lines 72: Citation?
- line 83: what are the criteria for qualification?
- line 87: not everyone is familiar with the festival and would not be able to imagine how this game was installed at the festival for collecting the data analyzed in this work.
- line 115: incomplete sentence due to typo, likely missing "asks" between "system" and "one."
- What was the shortcoming of the work described in line 122-126? Did they exhibit the issues of results being noisy due to variance in participants' effort and gullibility?
- The mention of the Chinese Room argument in line 147 is abrupt. What role does this serve in the narrative of the shortcoming of previous work?
- line 174-175: Is there any evidence that time constraints were encouraging System 1 judgements over System 2 reasoning? Also, is there evidence that System 2 reasoning is better than System 1 judgement for discerning between LLM and humans? Otherwise, this would be an unsubstantiated claim.
- What is the reverse effect of the Turing Test mentioned in line 191-192?
- line 199-207: doesn't this cause a downward spiral for initially underperforming players as they continue to get paired with underperforming players as they keep losing points due to the other players' mistakes? Shouldn't each player be scored based on their own prediction only? This may disincentivize collaborative interactions for gaining points but if the goal is to find the most proficient ones, this may be the better design.
- Is the formulation in line 216-221 correct in that the sigmoid is setup so that a bigger weight is assigned for games that were played earlier and lower weight is assigned for later games? (as k approaches 100, it gets closer to 0.5 while starting from closer to 1 when k=0) This seems counterintuitive since line 215 mentions measuring current strength, so shouldn't more recent games be weighted more?
- line 226: Why is a strong prior towards $S_i=1$ needed?
- line 285-286: typo? likely "the snips were taken once the game finished..."
- line 284, line 340-342: What are AllTalker, MadTalker, and MetaSim?
- What is the dialogue manager of the chatbots? Based on Figure 4, it is unclear how the bot decides to produce a turn given a three-way conversation. For instance, on the right side of Figure 4, Purple waits for others' responses but none is given by the bot, while in certain turns it produces two utterances at a time.
- line 336-337: I think examining the nature of games surrendered is going to be important, especially since it takes up a significant portion of games that were played.
- Section 4.3: This setup is akin to detecting LLMs, and therefore I would like to see how this result compares to DetectGPT, etc.
- Table 1: How is equation 8 applied to compute the weighted win ratio?

---

### Note · Authors · 2025-01-07

I have read and agree with the venue's withdrawal policy on behalf of myself and my co-authors.